# Key to Life: Physiological Role and Clinical Implications of Progesterone

**DOI:** 10.3390/ijms222011039

**Published:** 2021-10-13

**Authors:** Bernadett Nagy, Júlia Szekeres-Barthó, Gábor L. Kovács, Endre Sulyok, Bálint Farkas, Ákos Várnagy, Viola Vértes, Kálmán Kovács, József Bódis

**Affiliations:** 1National Laboratory for Human Reproduction, University of Pécs, 7624 Pécs, Hungary; szekeres.julia@pte.hu (J.S.-B.); kovacs.l.gabor@pte.hu (G.L.K.); esulyok@t-online.hu (E.S.); farkas.balint@pte.hu (B.F.); varnagy.akos@pte.hu (Á.V.); vertes.viola@gmail.com (V.V.); kalman.kovacs72@gmail.com (K.K.); bodis.jozsef@pte.hu (J.B.); 2Department of Obstetrics and Gynecology, Medical School, University of Pécs, 7624 Pécs, Hungary; 3MTA-PTE Human Reproduction Scientific Research Group, University of Pécs, 7624 Pécs, Hungary; 4Szentágothai Research Center, University of Pécs, 7624 Pécs, Hungary; 5Department of Medical Biology, Medical School, University of Pécs, 7624 Pécs, Hungary; 6Department of Laboratory Medicine, Medical School, University of Pécs, 7624 Pécs, Hungary; 7Doctoral School of Health Sciences, Faculty of Health Sciences, University of Pécs, 7624 Pécs, Hungary

**Keywords:** progesterone, progestogens, progestins, steroid, in vitro fertilization (IVF), intracytoplasmic sperm injection (ICSI)

## Abstract

The most recent studies of progesterone research provide remarkable insights into the physiological role and clinical importance of this hormone. Although the name progesterone itself means “promoting gestation”, this steroid hormone is far more than a gestational agent. Progesterone is recognized as a key physiological component of not only the menstrual cycle and pregnancy but also as an essential steroidogenic precursor of other gonadal and non-gonadal hormones such as aldosterone, cortisol, estradiol, and testosterone. Based on current findings, progesterone and novel progesterone-based drugs have many important functions, including contraception, treatment of dysfunctional uterine bleeding, immune response, and prevention of cancer. Considering the above, reproduction and life are not possible without progesterone; thus, a better understanding of this essential molecule could enable safe and effective use of this hormone in many clinical conditions.

## 1. Introduction

Progesterone is a key physiological component of the menstrual cycle, reproduction, and steroid hormone biosynthesis. Other physiological actions of progesterone in the central nervous system and immune system also support the concept that progesterone is key to life, and a better understanding of this important hormone helps its extensive clinical implication for human health. Progesterone was discovered because of its effect on the growth and implantation of embryos and extracted from the corpus luteum in the 1920s [1,2]. The history of progesterone research includes several milestones, and the exciting story is not yet over (Table 1). Progesterone also plays an important role in mammary gland development and affects the function of the central nervous system and cardiovascular system. In this review, we attempt to summarize the most important roles and clinical implications of progesterone (Figure 1). The most recent studies of progesterone research provide remarkable insights into the physiological and clinical importance of this hormone.

## 2. The Role of Progesterone in the Menstrual Cycle, Pregnancy, and Lactation

### 2.1. Preovulatory and Ovulatory Functions

The preovulatory follicles synthesize progesterone and also convert progesterone to estrogens [3]. FSH and LH act one after the other on ovaries enhancing steroid production; however, other factors can also be important in this “switch over” of steroid production. Acetylcholine and serotonin induce progesterone release from granulosa cells, while noradrenaline and nicotine significantly inhibit progesterone production [4,5,6,7,8,9,10].

Progesterone acts directly on granulosa cells by promoting follicular growth and inhibiting apoptotic genes via progesterone receptor membrane component-1 (PGRMC1) [3,11]. This non-genomic progesterone receptor is localized in the cell membrane. The modest increase in progesterone level appears to be a trigger of LH surge and the consequent ovulation. Progesterone increases GnRH release and enhanced gonadotropin pituitary sensitivity to GnRH.

Before ovulation, the granulosa cells of the dominant follicle begin their transformation into large luteal cells by becoming vacuolated and taking up the yellow pigment lutein [12,13]. Progesterone production of luteal cells depends on the availability of circulating cholesterol substrate and is facilitated by a low-level LH stimulation. Large luteal cells have a greater steroidogenic capacity but lack the LH and human chorionic gonadotropin (hCG) receptors. The small luteal cells, likely to be derived from the theca cells, contain LH and hCG receptors and are linked to large cells by gap junctions. With the help of rapid transport of signals between the cells, large luteal cells respond to LH stimulation and synthesize progesterone.

### 2.2. Progesterone in Premenstrual Syndrome

Premenstrual syndrome (PMS) is a common condition affecting about 3–8% of menstruating women. It presents with a quite wide range of psychological and physical symptoms severe enough to limit the patients’ everyday activities. During the last decades, particular attention has been paid to explore the association of hormonal dysregulation with depression, cognitive functioning, anxiety disorders, as well as with fluid retention. The pathophysiology of PMS is complex, imprecise, and not fully understood, although there is some evidence to support the possible role of progesterone [14].

Progesterone freely passes the blood-brain barrier, is metabolized in the brain to allopregnanolone and pregnanolone, which, by stimulating the gamma-aminobutyric acid (GABA) inhibitory system, mediates some of the neuropsychiatric symptoms of PMS. On the other hand, progesterone interacts with the central serotoninergic system by reducing the bioavailability of serotonin through the activation of monoamine oxidase [14,15].

Importantly, progesterone has been shown to attenuate the inflammation-induced activation of indoleamine-2, 3-dioxygenase (IDO) and to reduce tryptophan catabolism to kynurenine and to neurotoxic metabolites, all relevant for neuro-inflammatory diseases. As a consequence, IDO inhibition helps to maintain the cellular tryptophan pool and to direct tryptophan catabolism toward the serotonin pathway. As some women with PMS have low or decreasing progesterone levels or reduced responsiveness to this hormone, progesterone replacement or selective serotonin reuptake inhibitor administration appears to be feasible in selected cases [16,17].

The involvement of prolactin in the pathogenesis of PMS is a matter of debate. Decades ago, elevated prolactin levels were shown to be associated with symptoms of PMS that improved after the suppression of prolactin secretion with bromocriptine. Prolactin was assumed to cause PMS by enhancing renal salts and water retention or by interacting with ovarian hormones and/or lithium-specific brain structures [18,19]. This concept, however, has been challenged as conflicting experimental and clinical data have been accumulating.

### 2.3. Progesterone in the Luteal Phase

Progesterone is produced by the corpus luteum, and it is the dominant hormone after ovulation in the luteal phase. In the early luteal phase, progesterone secretion is stable and does not correlate to LH pulses, while in the mid- and late luteal phase, progesterone secretion is episodic and correlates with the pulsatile release of LH. During this period, a progressive reduction in LH pulse frequency and amplitude occurs [20]). Compared to low levels (1–2 nmol/L) during the follicular phase, progesterone levels increase to 15–20 nmol/L in the early luteal phase and then peak in the middle of the luteal phase (35–50 nmol/L). If conception does not occur, the corpus luteum starts to break down 9 to 10 days after ovulation, causing progesterone levels to fall (20–40 nmol/L). Lowered plasma steady-state levels of progesterone combined with declining progesterone levels during the luteal phase may predict premenstrual syndrome (PMS); however, some clinical trials did not show that progesterone is an effective treatment for PMS [15,21].

In addition to progesterone, the human corpus luteum secretes relaxin and oxytocin, which may function as paracrine or autocrine modulators of luteal function [22].

### 2.4. The Endometrial Effect of Progesterone

Rising progesterone levels in the early luteal phase play an important role in the transition of the endometrium from the proliferative to the secretory phase. Because of the suppressive action of progesterone during the ovulatory phase, estrogen receptors of stromal and myometrial epithelial cells drop rapidly. At the same time, due to estradiol action, the expression of progesterone receptors (PR) exponentially increases in endometrial endothelial cells and then dramatically decreases in the late luteal phase [3,23]. Endometrial progesterone and its metabolite 17α-hydroxyprogesterone are significantly associated with endometrial receptivity [24]. A higher proportion of receptive endometria was observed when endometrial progesterone levels were higher than 40.07 µg/mL, and a lower proportion of receptive endometria was associated with endometrial 17α-hydroxyprogesterone lower than 0.35 ng/mL.

### 2.5. Progesterone during Pregnancy

Progesterone is recognized as a key physiological component of implantation and maintenance of pregnancy. Removal of corpus luteum or treatment with progesterone antagonist, e.g., mifepristone, terminates the pregnancy [25,26].

Progesterone is crucial for preparing the endometrium for implantation and for regulating trophoblast invasion and migration. Progesterone establishes uterine receptivity by blocking the proliferative effect of estrogen, by inducing genes that allow the endometrium to permit embryo attachment, and also acts as a negative regulator of trophoblast invasion by controlling matrix metalloproteinase (MMP) activity [27].

Both genomic and non-genomic actions of progesterone play a key role in adequate decidualization and implantation. In addition to their genomic actions mediated by the classical nuclear progesterone receptors (PR-A and PR-B), progesterone may also trigger rapid cytoplasmic signaling events. Membrane-bound PRs have been implicated in the rapid non-genomic actions of progesterone [27].

In human pregnancy, progesterone production is eventually sustained by the placenta, and the serum concentrations of progesterone range from about 100 to 500 nmol/L. Progesterone is crucial not only for the establishment but also for the maintenance of pregnancy, both by its endocrine and immunological effects [28,29]. The maternal tolerance of the fetal “semi allograft” is critical for a successful pregnancy. The maternal immune system recognizes the fetal antigens, thereby resulting in lymphocyte activation and induction of PRs in immune cells. The immunomodulatory effect of progesterone is mediated by a protein named the progesterone-induced blocking factor (PIBF) [30]. In the presence of progesterone, PR-positive lymphocytes produce this mediator protein. PIBF expression of the lymphocytes shows an inverse correlation with NK activity.

PIBF increases the production of Th2 cytokines and inhibits the degranulation of NK cells. Lymphocytes of women with threatened abortions fail to produce this factor. Other studies showed that stress-induced abortion in mice significantly reduced progesterone- and PIBF levels. The percentage of PIBF-positive lymphocytes in the peripheral blood of healthy pregnant women is significantly higher than in that of women at risk for premature pregnancy termination [30].

### 2.6. Progesterone and Tryptophan Catabolism

Progesterone has also been claimed to play a prominent role in the control of tryptophan catabolism that provides essential compounds to achieve successful pregnancy. The essential amino acid tryptophan acts as a precursor to various metabolic pathways, including protein synthesis and production of serotonin and kynurenine. The kynurenine pathway represents the major non-protein route for tryptophan metabolism (>95%). Along this pathway, bioactive intermediates are generated that are involved in pregnancy-related immune tolerance, inflammation, oxidative stress, neuroprotection/toxicity.

Two enzymes initiate tryptophan catabolism; the hepatic tryptophan–2, 3-dioxygenase (TDO) that is induced by glucocorticoids and tryptophan and inhibited by progesterone and oestrogens, as well as the extrahepatic indoleamine-2, 3-dioxygenase (IDO) that is up-regulated by inflammatory cytokines, in particular, by interferon gamma [31] (Figure 2).

The mRNA and protein expression for both TDO and IDO has been documented in the placenta, decidua, and early embryonic/fetal tissues. Importantly, progesterone proved to inhibit IDO mRNA expression and enzyme activity. Reduced conversion of tryptophan to kynurenine, therefore, results in an increase in the bioavailability of tryptophan for protein and serotonin synthesis [32].

Convincing evidence has been provided for the essential role of tryptophan catabolism to both kynurenine and serotonin in implantation, early embryonic and fetal development. Therefore, the two metabolic routes should be kept in balance, or even a slight serotonin dominance is required [33]; progesterone appears to be involved in the regulation of these delicate processes.

### 2.7. Progesterone in Follicular Fluid

In menstrual cycles, steroid concentration of follicular fluid increases during the follicular phase until the onset of LH surge. From the LH surge to ovulation, progesterone concentration increases dramatically, whereas 17β-oestradiol level decreases [34,35,36,37].

After the LH surge, progesterone is the dominant steroid in preovulatory follicular fluid [38,39], which is involved in normal ovarian function and may play an important role in oocyte maturation, ovulation, embryo development, implantation, and maintenance of pregnancy. Steroids are essential to normal meiosis and fertilization, and blastocyst development [37,38,40]. Progesterone plays a role in the regulation of different biological functions, such as the resumption of meiosis, fertilization, implantation, and embryonic development [41,42,43].

Steroidogenesis is essential for follicular growth synchronization and oocyte development [44]. Follicular fluid containing progesterone is critical for oocyte health [44]). Follicular fluid plays a role in determining oocyte quality and developmental potential to achieve fertilization [45,46]. Ovarian cellular cholesterol is converted to pregnenolone, and then 3-beta-hydroxysteroid dehydrogenase converts pregnenolone to progesterone [47]). Although biosynthesis is well known, the mechanism of the effect of progesterone on oocytes is not completely understood. Despite the fact that the biological actions of progesterone are usually mediated by PRs, these receptors have not been found on the oocyte [38]. However, PR mRNA and proteins have been identified in mature cumulus-oocyte complexes and embryonic cells [48,49,50,51,52]. Progesterone receptor membrane component 1 (PGRMC1) has been detected in human oocytes [53], which is also a potential mediator of progesterone action. Molecules and protein pathways working in human follicular fluid may affect follicle growth and oocyte maturation [54].

In vitro fertilization is the most advanced medical technology for the treatment of infertility. Although the effectiveness of the methods, as well as the physiological and pathological knowledge, has increased, the quality and quantity of embryos and the success rate of conventional in vitro fertilization (IVF) and intracytoplasmic sperm injection (ICSI) procedures are still not satisfactory. The success of assisted reproduction depends on several factors, many of which remain unclear. Follicular steroids are implicated in the maturation of the oocytes, which requires the resumption of meiosis and achievement of cytoplasmic maturation [36,55,56]. Considering the above, it is reasonable to assume that progesterone could help or at least indicate a normal fertilization potential.

The results of several studies correlating follicular steroid concentrations with subsequent oocyte fertilization and embryo development suggest that the follicular fluid environment may play a role in oocyte competence to fertilization and embryo development during conventional IVF [36,44,46,57,58,59] and ICSI [46,60,61]. Our recent meta-analysis has revealed that fertilized oocytes were derived more often from follicles containing significantly higher levels of progesterone than unfertilized oocytes in the case of both conventional IVF and ICSI [37]. Currently, it is not possible to find a relationship between follicular steroid hormones and morphological grades of embryo quality [60,62]. The steroid content of follicular fluid may change earlier than morphological signs of oocyte degeneration appear [36,63].

### 2.8. Progesterone and Lactation

Progesterone and progesterone receptor B have primary importance in mammary gland development by enhancing epithelial cell proliferation and differentiation. These effects are achieved through acting in concert with insulin-like growth factor-1 (IGF-1). As a result of their synergistic action, ductal length increases, and an extensive network of branching develops. Furthermore, progesterone increases the anti-apoptotic effects of IGF-1 and induces alveolar development when IGF-1 and oestrogens are present [64]. Progesterone also serves as an inhibitor of lactogenesis during gestation, and the postpartum decrease in progesterone levels is required to ensure full lactation. If plasma progesterone remains elevated or progesterone is sequestered into the adipose tissue of the mammary glands, lactogenesis is delayed, and lactation failure may occur [65].

Concern has been expressed about the effects of progesterone-containing contraceptive pills when taken by breastfeeding mothers. Meta-analyses have clearly demonstrated, however, that during established lactation, there were no differences in progesterone levels, milk yield or composition, and duration of breastfeeding when mothers with or without contraceptives were compared [66].

## 3. Clinical Implications of Progesterone

Progesterone and other progestational agents have several clinical applications, and extensive research has been conducted to evaluate the physiological effects and also the side effects of exogenous progesterone administration [67]. Progesterone is the only native ligand [68], while progestogens comprise all substances that activate PR and result in progesterone-like effects, and progestins are synthetic PR agonists [69].

After oral administration, more than 90% of progesterone is metabolized during the first hepatic pass [70], leading to decreased efficacy and also high levels of metabolites that may be responsible for dizziness and drowsiness. Because of the poor oral absorption and rapid first-pass metabolism of oral progesterone, a variety of oral, injectable, and implantable synthetic analogs have been developed [71]. Progestins, e.g., medroxyprogesterone acetate and norethindrone acetate, were designed to resist enzymatic degradation and remain active after oral administration. However, synthetic progestins may exert significant side effects, such as dysphoria, depression, anxiety, fatigue, as well as headaches, hypercoagulant states, increased androgenicity, reduction in HDL cholesterol levels, and fluid retention [70]. Transdermal administration of norethindrone acetate can also exert undesirable effects on the liver.

In the case of transvaginal administration of progesterone with the help of suppositories or with polycarbophil-based gel, the plasma progesterone concentration remains low, resulting in minimal side effects. However, the local vagina to uterus transport of progesterone results in uterine uptake of progesterone and allows secretory transformation of the endometrium and maintenance of pregnancy despite the low plasma progesterone levels [70].

Subcutaneous administration of progesterone is effective and safe. No statistically significant or clinically significant differences exist between subcutaneous and vaginal progesterone administration for luteal phase support [72].

Natural progesterone is obtained primarily from plants such as soybeans and Mexican yam roots [71]. Micronization decreases the particle size and enhances the dissolution, resulting in an increased half-life and decreased destruction in the gastrointestinal tract. An oral micronized progesterone has improved bioavailability and fewer side effects compared with synthetic progestins [71]. Micronized progesterone has been shown not to affect mood or decrease HDL cholesterol levels; however, fatigue and somnolence have still been reported as side effects. Therefore oral administration of micronized natural progesterone appears to be a safe and effective alternative to synthetic progestins [67].

### 3.1. Applications of Progesterone in Reproductive Medicine

#### 3.1.1. Progesterone Treatment in Assisted Reproduction

Progesterone seems to be the best option for luteal phase support in assisted reproduction cycles, with better results when synthetic progesterone is used [73]. Meta-analysis proved that the supplementation of the luteal phase with vaginal progesterone significantly increases live birth among women undergoing intrauterine insemination when receiving gonadotropins for ovulation induction [74]. Women stimulated with clomiphene citrate to induce ovulation do not seem to benefit from this treatment. Progesterone administration can be initiated at the time of oocyte retrieval, or 1–2 days before embryo transfer, and continued until the first positive hCG test or until the onset of menstruation [75,76].

Although progesterone supplementation beyond the first positive pregnancy test after IVF/ICSI might generally be unnecessary [77], most IVF centers have elected to continue progesterone administration until the 6th–10th week of pregnancy to minimize the risk of miscarriage [75].

#### 3.1.2. Progesterone in Recurrent Miscarriage

Women with a history of miscarriage who present with bleeding in early pregnancy may benefit from the use of vaginal progesterone. Treatment with vaginal micronized progesterone 400 mg twice daily was associated with increased live birth rates [78].

#### 3.1.3. Maintenance of Uterine Quiescence in Late Pregnancy

Progesterone administration maintains uterine quiescence in late pregnancy and delays delivery, particularly in patients with short uterine cervixes [75]. Recent meta-analysis proved that vaginal progesterone decreases the risk of preterm birth and improves perinatal outcomes in singleton gestations with a mid-trimester sonographic cervical length ≤25 mm [79].

#### 3.1.4. Progesterone and Endometriosis

It has been shown that women with endometriosis-related infertility may be partially resistant to progesterone actions on the endometrium [80,81]. In endometriotic stromal cells, the levels of PR, particularly the PR-B isoform, are significantly decreased, leading to a loss of paracrine signaling. Another possible mechanism of progesterone resistance in endometriosis is a deficiency of co-regulator Hic-5, which binds to steroid receptors and modifies their nuclear effects. Alteration of expression or function of progesterone receptor chaperones and co-chaperones is also a potential modifier of progesterone resistance. Although simply inducing peritoneal lesions can result in alterations in progesterone action, it is presumed that local inflammation is involved in progesterone resistance in endometriosis. Progesterone induces the expression of 17beta-hydroxysteroid dehydrogenase type 2 (HSD17B2), an enzyme that metabolizes estradiol to less potent estrone. In the case of PR resistance, there is increased estrogen bioactivity upon loss of HSD17B2 function, which in turn induces an inflammatory response in the endometrial tissue characterized by an elevated level of inflammatory cytokines [82]. Women with endometriosis-related infertility might achieve normal endometrial secretory-phase function and structure with higher progesterone doses or with treatments targeted at abnormal inflammation [80]. It seems that some of the defects of progesterone actions are likely to be overcome with higher concentrations of progesterone.

### 3.2. Birth Control

In combined oral contraceptive pills, the doses of estrogen are not sufficient to prevent ovulation. However, the progestin component, or in the case of minipills, progestin alone, suppresses GnRH, leading to decreased FSH and LH release [71,83]. The estrogen component potentiates the action of progestin and stabilizes the endometrium reducing breakthrough bleeding. Progestins may prevent implantation because of their atrophic effect on the endometrium. Moreover, progestins minimize sperm penetration by enhancing the development of thick cervical mucus.

Several progestins were developed to achieve a suitable contraceptive effect with minimal risk of side effects (Figure 3). Although the first generation of 19-nortestosterone derivatives has androgenic side effects, later potent antiandrogenic progestins have become available. The most potent antiandrogenic progestins are the following: cyproteron acetate, dienogest, drospirenone, chlormadinone acetate. These molecules are antagonists of 5α reductase, and they also block testosterone from binding androgen receptors leading to decreased androgen effect.

Drospirenone, as an analog of spironolactone, has anti-mineralocorticoid and anti-andogenic effects [84]. It also counteracts the estrogen-induced stimulation of the renin-angiotensin-aldosterone system. Because of these characteristics, drospirenone has the potential to reduce blood pressure and low-density lipoprotein levels and to enhance high-density lipoprotein levels. It has negative effects on water retention and body weight leading to favorable body weight control [84,85].

Besides oral contraceptives, intramuscular injections of progestins, subcutaneous levonorgestrel implant, and the levonorgestrel IUD provide a high degree of contraceptive efficacy [86]. Postcoital or emergency contraception with levonorgestrel can prevent about 75 percent of pregnancies that would occur without treatment [71].

#### Application of Progestational Agents in Other Gynecological Conditions

Progestational agents with or without estrogen are often used to correct irregular bleeding [71]. Progesterone and progestins have also been used in the treatment of gynecological conditions, including dysfunctional uterine bleeding, oligomenorrhea, polymenorrhea, hypermenorrhea, dysmenorrhea, secondary amenorrhea, endometriosis, and in menopausal hormone therapy. Women with polycystic ovary syndrome, adolescents, and perimenopausal women may require progestational agents for dysfunctional uterine bleeding resulting from anovulatory cycles [71,87].

Progestational agents also protect against endometrial cancer, endometriosis, premenstrual tension, and dysmenorrhea [86].

### 3.3. Hormone Replacement Therapy

Conventional hormone replacement therapy (HRT) includes estrogen and progesterone treatment to mimic hormones created by the human ovary [88]. Hormone replacement can be used in women with premature ovarian failure, menopause, and secondary (pituitary) or tertiary (hypothalamic) hypogonadism [89]. Progestins are given with estrogen to stimulate puberty and secondary sexual characteristics in women with primary ovarian failure, for example, Turner syndrome [89].

Progestins, along with estrogen, can be used in postmenopausal women with an intact uterus for hormone replacement therapy to relieve the symptoms associated with menopause (vasomotor symptoms or genitourinary symptoms) and to prevent osteoporosis [88,89]. Estrogen alone will cause the endometrial lining to grow, so a woman with an intact uterus must have progestogen with estrogen to protect her uterus from endometrial hyperplasia or malignancy [88]. When the use of progesterone is necessary, micronized progesterone is considered the safer alternative [90]. Progesterone can provide symptom relief from sleep disturbance and mood instability and may offer tissue protection to the breast [88]. The long-term use of progesterone therapy is associated with a lower risk of developing high-risk serrated polyps [91].

### 3.4. Oncological Aspects of Progesterone

The use of progesterone has been linked to lower rates of uterine and colon cancers and may also be useful in treating endometrial carcinoma, ovarian carcinoma, melanoma, mesothelioma, and prostate tumors [92].

Although the results of studies discussing the role of progesterone in breast cancer are controversial, a recent meta-analysis involving 86,881 postmenopausal women reported that the use of natural progesterone was associated with a significantly lower risk of breast cancer compared with that of synthetic progestins [92]. These results support the view that natural progesterone does not cause breast cancer. Selective progesterone receptor modulators have significant activity against breast cancer in clinical trials [93].

### 3.5. Progesterone and Nervous System

Progesterone is synthesized by neurons and also by glial cells, and this neurosteroid is involved in the regulation of various molecular and cellular processes underlying neurogenesis, myelination, neuroprotection, neuromodulation, learning, memory, and mood [94].

Clinical trials showed that progesterone infusion after traumatic brain injury results in reduced neuronal loss, enhanced remyelination, improved functional recovery, and an overall decrease in cerebral edema [95].

## 4. Summary and Conclusions

Progesterone is necessary for successful embryo implantation and pregnancy maintenance. Vaginal progesterone treatment minimizes the risk of recurrent miscarriage and decreases the risk of preterm birth, saving many fetal lives.

However, progesterone is far more than a gestational agent (Figure 4). Progesterone is an essential steroidogenetic precursor of other gonadal and non-gonadal hormones such as aldosterone, cortisol, estradiol and testosterone. These hormones are responsible for innumerable functions such as sodium conservation in the kidney, regulation of blood pressure, response to stress and low blood-glucose concentration, development of female and male secondary sexual characteristics. Progesterone also plays an important role in the nervous system. Its neurogenic effect is essential for normal brain development in fetuses, while the neuroprotective effect of progesterone improves the patient’s survival after traumatic brain injury. Progesterone and novel progestins have many important functions, including contraception, luteal phase support, treatment of dysfunctional uterine bleeding, and endometriosis. Progesterone has an important role in immune response and also in the prevention and treatment of various cancers.

Although several studies prove the importance of progesterone in various essential physiological processes, we are far from completely understanding the key role of progesterone in the miracle of life.

## Figures and Tables

**Figure 1 ijms-22-11039-f001:**
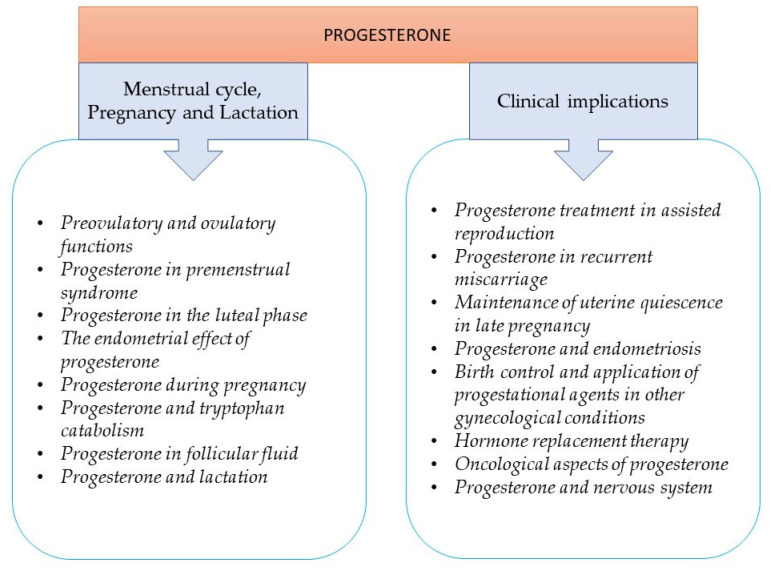
Contents of the review.

**Figure 2 ijms-22-11039-f002:**
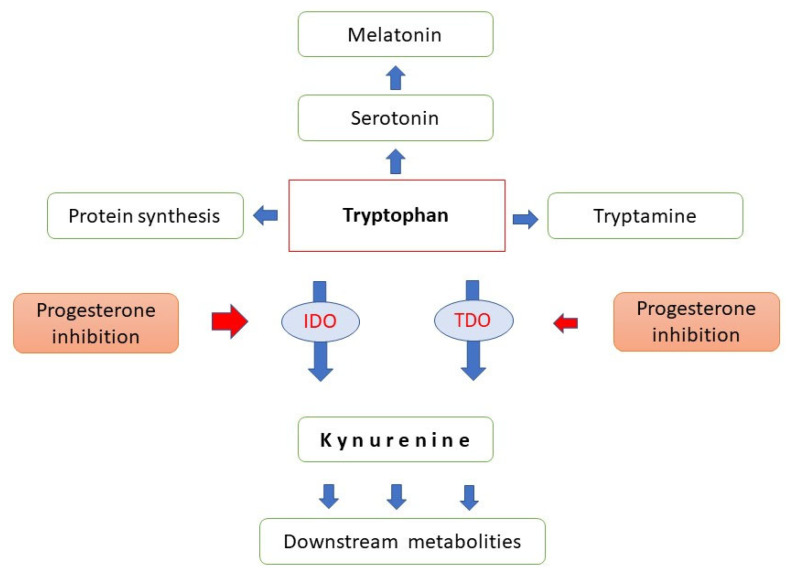
Major pathways of tryptophan metabolism. IDO: indoleamine-2,-3- dioxygenase, TDO: tryptophan-2,-3-dioxygenase, red arrows indicate progesterone inhibition.

**Figure 3 ijms-22-11039-f003:**
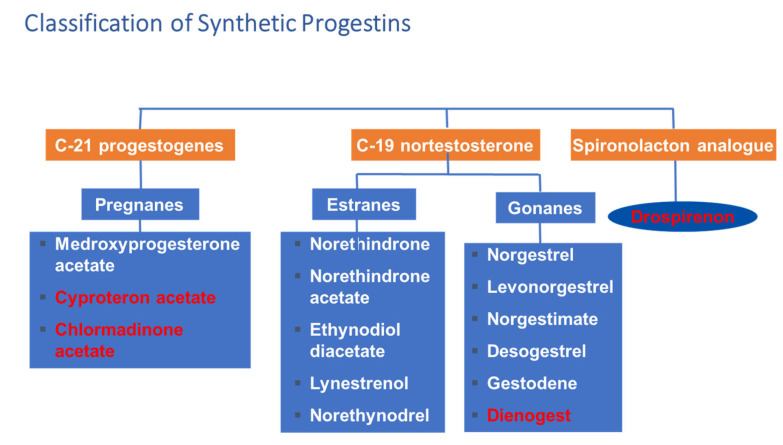
Classification of synthetic progestins. Red color indicates antiandrogenic effect.

**Figure 4 ijms-22-11039-f004:**
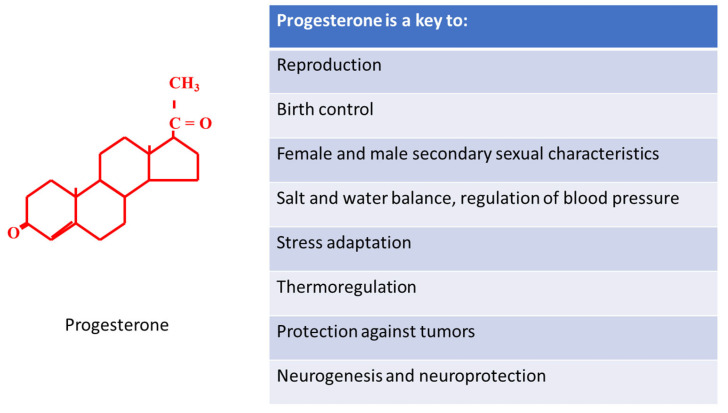
Is progesterone really a key to life?

**Table 1 ijms-22-11039-t001:** Major milestones of progesterone research.

1897	J Beard	Absence of ovulation during pregnancy
1898	L A Prenant	Corpus luteum is necessary for implantation
1916	E Herrmann and M Stein	Extract from the corpus luteum blocks the ovulation
1923	W M Allen	Effect of the corpus luteum extract on castrated rabbit’s endometria
1931	L Haberlandt	Birth control through temporary hormonal sterilization
1933–1934	K H Slotta, P Wintersteiner, A Butenandt, U Westphal, W M Allen, M Hartmann, A Wettstein	Isolation of progesterone
1951	C Djerassi	First synthesis of an oral contraceptive norethindrone
1970	B W O’Malley	Progesterone receptors and isoforms

## Data Availability

Not applicable.

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
