# Peer review of "Key to Life: Physiological Role and Clinical Implications of Progesterone"

_ijms, 2021, doi:10.3390/ijms222011039_

Round 1
Reviewer 1 Report
this is an excellent review of a presumably well known substance. Tables and figures summarising details would be helpful
Author Response
Thank you for the time and effort that you have dedicated to providing your valuable feedback on our manuscript.
Comments from Reviewer 1
- this is an excellent review of a presumably well known substance. Tables and figures summarising details would be helpful
Thank you for your valuable comment. We added two new figures (Figure 1. and Figure 2.) to summarize details of the review.
Reviewer 2 Report
This is a well written review on the function of progesterone in reproduction as well as clinical uses. It cites many recent manuscripts. The initial sentence of the abstract should be changed, other than that i had no other suggestions.
figure 1 could be included in the text and is probably not necessary for the manuscript. If the authors want to retain figure 1, perhaps they could reformat it into a table which would make more sense.
Author Response
Thank you for reviewing our manuscript and for your valuable comments. Thank you for the time and effort that you have dedicated to providing your feedback.
- This is a well written review on the function of progesterone in reproduction as well as clinical uses. It cites many recent manuscripts. The initial sentence of the abstract should be changed, other than that i had no other suggestions.
We have changed the initial sentence of the abstract: “The most recent studies of progesterone research provide remarkable insights into the physiological role and clinical importance of this hormone.”
- figure 1 could be included in the text and is probably not necessary for the manuscript. If the authors want to retain figure 1, perhaps they could reformat it into a table which would make more sense.
Thank you for your comment. We have reformatted the Figure 1 into a table (Table 1).